# Chatbot: a Key for Understanding AI Revolution

Guido Tascini
ISAR-Lab Study and Research Center,
Former Director of UNIVPM AI Lab
guido.tascini@gmail.com.

*Abstract*--**One way that clearly shows how an intelligent machine works is to design a chatbot to show how man-machine collaboration can take place. Here AI is used to teach a man how to drive a car . The intelligent system is equipped with a specially designed chatbot. With it the system can respond to the driver's requests. At the same time he learns the behavior of the driver: both from the questions this makes through the chatbot, and from how he performs the various driving maneuvers. It is understood that the system is equipped with a set of sensors and adequate software to ensure the safety of the driver and other vehicles involved in his driving. In extreme cases, of danger or serious non-compliance with road regulations, the system slows down and can come to a standstill. Created chatbot is able to converse with the driver, that is an elderly subject to problems typical of old age. Assistive software robots are important while allows to understand how the AI works by providing support and companionship. A software Chatbot, able to understand natural language and to learn from interactions, active 24 hours a day, provides answers to user questions. Fields supplying matter to implement such a bot are: *Machine Learning*, S*peech Recognition*, *NLP*, and C*hatbot Training* with commercial platforms.**

**Keywords - Explanable AI, Chatbot, AI Understanding, Deep Learning, Speech Recognition, NLP.**

## I.Introduction

The *Understandable Artificial Intelligence* (XAI) is working its way towards making AI more understandable and clear in contexts where the lack of transparency is viewed with suspicion.

The current relaunch of AI in applications requires the clarification of the interaction process between man and intelligent machine, by establishing integrated ways of working, in which human beings and intelligent problem-solving systems cooperate.

On the other hand, cooperation[37] is important in consideration of the trust that people place in intelligent machines and increasingly entrust to these responsibilities and skills. To do this, people need to *clearly* understand how the smart machine behaves , since the algorithms that drive it are not at all.

So an approach based on dialogue between humans,

perhaps the elderly, and the intelligent machine seems the key to this understanding and to overcoming the reservations and doubts that people have towards AI.

The fact that man is an elder shows an important aspect of intelligent machines: being able to adapt to a problematic person, learning his behavior from experience. And this is a typical aspect of intelligent learning systems.

The *Chatbots* are Artificial Intelligence software systems, capable to create a conversation between a virtual agent and a user. Many complex features that such a system must possess are difficult to achieve: human language understanding and learning from interactions, by Gradually increasing one's knowledge; commitments remembering; remotely connecting, physiological parameters controlling, human entertaining. The *Assistive intelligent software robots* are a demonstration of what an AI application is capable of doing.

These software robots (Chatbots), answering the user's questions and then educating them on everything: on the guide and on their behavior, for the non-specialists are much more understandable of the Deep Learning algorithms, which require careful interpretation.

These architectures are composed of multiple levels with non-linear operations, such as in neural nets with many hidden layers. Learning algorithms, such as those for Deep Belief Networks (DBN), that Geoffrey E. Hinton[3] et al. have proposed have achieved a remarkable success. The reason for the adoption of machine learning is related to the distance that, up to half of 2000, existed between AI systems and human brain's ability to solve problems of: vision, natural language understanding and speech recognition.

The *inspiration to human brain* suggested to design machine learning with properties that are evident in its visual cortex, like its deep and layered connectivity. Yet the attempt to train neural networks with more than one hidden layer has failed until mid-2000.That is the performances was not superior to those of non deep, (*shellow*) networks. In 2006, Geoffrey Hinton[3] et al. designed the *deep belief network*: a probabilistic neural network with an efficient 'greedy' procedure for successfully pre-training it. The procedure is linked to the learning algorithm of the restricted Boltzmann machine (RBM) for layer-wise training of the hidden layers, in an unsupervised fashion. Later the procedure was generalized (Yoshua Bengio[5] et al.) and subsequently many other works appeared which have strengthened the field of deep learning[7 8 9 10 11 12 13] Main

characteristics of deep learning networks processing show their execution complexity:

- Each layer is pre-trained by unsupervised learning .
- The unsupervised learning of representations is used to (pre-)train each layer.
- The unsupervised training of one layer at a time, is on top of the previously trained ones. The representation learned at each level is the input for the next layer.
- The supervised training is used to fine-tune all the layers (in addition to one or more additional layers that are dedicated to producing predictions).

## II. Chatbot Characteristics

Chatbot means chatting robot. These are automated systems, programmed to respond to certain inputs and resulting user feedback. They currently operate mainly in chat. For example, on Facebook Messenger, it counts a lot. It is a software based on Artificial Intelligence, which simulates a smart conversation with the user on a chat. In practice, they are currently planning to offer a functional and support service through the major messaging platforms.

A conversation with a chatbot is very easy and a mean to increase our proper knowledge in every field, mainly on the world of AI. This stolen knowledge is obtained in a much easier way than by following the behavior, layer by layer, of a deep neural network.

Another main idea is to destine the AI-chatbot to be a companion for senior people. The problem here is to give senior people the opportunity to communicate by talking, do answers, share their experiences and memories.On the other side the chatbot would be able also to conduct a conversation on topics such as: weather, nature, news, history, cinema, music, etc.

But the special feature of a Chatbot is its ability to respond to question posed by the human.

This may pose questions on functioning of an AI System (that is on its functioning).

As an *assistant*, the AI Chatbot will be able to:
 - entertain the man with intelligent speeches, offering games, or delivering news.
 - ask and answer questions,
 - memorize the context of the conversation, for a useful dialogue[20 25 34 35].
 - Answer to driver on machine functioning.
 - Answer to elder question, in particular on Artificial Intelligence.
 - do raccomandations.

### Example of Chatbot functioning.

For instance if we pose a question: "*how did you slip?*",we can consider three answers:
*good*, *normal* or *bad*.
But if the interlocutor goes into details?
Of coarse for such cases, the chatbot has to be provided with 'words of support'. The solution now offered by different platforms may be an *API*, built to understand the context, synonyms or other situation. APIs may be supplied for *speech recognition issues*, and for much

more. In general, the chatbot's functions are various and can be defined only during the process of *chatbot developing and testing*.

Our Chat Bot, as intelligent software, has to be capable of learning from the experience and has to be implemented by introducing deep machine learning algorithms..

### Man-Machine Cooperation

It is fundamental, in order to understand how an AI System works, to show how a chatbot may supply knowledge to the people. Here we delegates to the chatbot the task of teaching the driving of the car : to this aim in particular we give to chatbot the opportunity of collaborate with human.

The intelligent system that aid, in various manners including the automotive task, the elderly people is equipped with a specially designed chatbot. This can answer to the most of the driver's questions.

At the same time the chatbot learns the behavior of the driver. It achieves this through both: from the questions he poses to the chatbot, and from how he performs the various driving maneuvers.

Through all this it is performed a man machine cooperation and Chatbot gradually learns from the experience the behavior of the driver; then tries to correct the wrong behavior of man.

Some examples suggest the chatbot's behavior towards the driver.

Example1. Man talks to the Chatbot and asks for help. Then machine answers: - "*what do you want to learn*?"

Man:- *I'd like to learn drive this car.*

Machine: - *put in motion.*

Man:- *ok. Done.*

Machine:- *enter Drive.*

Example2. Route control using a navigator. Man learns to *turn*. Man-Machine dialog.

Man: *turn right*

The Chatbot has the link to the motor and may control all its parameters.

*Situation a.*Overtaking occurs.

Machine (Chatbot):- *give way*

*Situation b.* Crossroad

Machine:- *give right to pass*

Machine:- *stop. Give precedence*

The passenger of coarse can ask the chatbot about his guide. The chatbot, on the other side, stores the passenger's behavior.

The knowledge so accumulated will be used by chatbot to advise the passenger when driving:
 - when he asks the chatbot,
 - and when is driving autonomously.

Example3. The driver need to brake.

Man: - *as a brake*?

Machine:- *Push right pedal.*

Example4.Man turns right (for navigator command)

Machine:- *Put the arrow* !

(Reccomandation due to the abitude of driver of not putting the arrow)

All the recommendations derive from the learning of the driver behavior.
Examples of situation in what the man may need recommendations:
- *turns*
- *Exchange*
- *precedence*
- *braking*
- *Arrest*
- *Parking area*
        *Etc.*
It is understood that the system is equipped with a set of sensors and adequate software to ensure the safety of the driver and of the other vehicles involved in his driving.
In extremis, in case of danger or serious non-compliance with road regulations, the system slows down the car that can come to a standstill.

## Understandable Artificial Intelligence (XAI)
The Understandable Artificial Intelligence is born to make AI more understandable in contexts where the lack of transparency is viewed with suspicion.
The current relaunch of AI applications requires the clarification of the interaction process between man and intelligent machine.
Integrated ways of working are being established, in which human beings cooperate with intelligent problem-solving systems[37].
The trust that man must place in intelligent machines is fundamental, given that to these are increasingly entrusted responsibilities and skills.
It is important to make man understand how the intelligent machine behaves in a clear way, given that the algorithms that guide it are not.
An approach based on dialogue of human, in particular if helderly,   with the intelligent machine seems the key to clear reservations and doubts towards AI.

## III.  Understanding NL with NN.

Giving machines the ability to learn and understand human language opens new scenarios   in AI applications. Natural Language Understanding in the recent past was based on complex systems; this has been overtaken by the recent approaches in terms of 'word representation, and processing'[32] with Deep Learning networks.

An example may explain better than many words: how a chatbot tries to understand a word. Suppose the word is: *Undefined.*
We try to gather information about it, such as definition, sentiment and more.
Let's try to break the word into three parts:
Prefix-stem-suffix.
Stem represents 'define', from which we deduce definition and feeling of the word. The prefix 'un' indicates opposition, the suffix 'ed' indicates past time.
Such an approach leads to an enormous amount of data: a lot of prefixes, suffixes and roots must be considered.
Then a different approach may be used: the machine

constructs a word map, containing also their meanings and their interactions with other words;
in practice it builds a dictionary by linking the words, semantically and contextually, to other words and contexts.
Each word is mapped to a set of numbers in a high-dimensional space, in which similar words are close, while the different words are distant[18][19][28][29].
The machine can learn all this, and learning varies depending on what the machine reads:
- a large amount of texts or
- texts relating to a particular task.
A progress was achieved by organizing the proximity of words to the last word of a pronounced sentence. The tab1 shows an example of this organization.

Tab 1.

| Expression | | Nearest Token |
|---|---|---|
| Palace - balcony windows | + | Apartment |
| Cellar- dehumidifier Photovoltaic cells | + | Roof |
| Department- product Office | + | Document |
| Sun - day night | + | Moon |

So the prediction of a word in a sentence can be obtained with a simple metric: given the sentence, every word in the dictionary has assigned the probability that the word appears later in the sentence.
For example, if we consider a sentence with a part to complete like this:
 "I'm driving ______"
If the candidate words are, for example, *tank* and *car*, car has a high probability of appearing after 'I'm driving', while tank has a very low probability. The probability value causes the word "auto" to be placed next to the word "driving" while putting the word "tank" away from it.
A next step in the chatbot in understanding the language is the *modeling of language*[2][4]:. An example is to group words into small sentences, called n-grams, grammatically correct, and that have sense. The modeling of the language uses n-grams-groups of words and processes them further with heuristic algorithms; finally it inserts them into automatic learning systems.
A Look Up Table contains the **Word Embeddings**.
It returns a vector given a word, and a matrix given a sentence.
We can create a vector of words[17], for each word by writing a recurrence matrix: it contains the number of counts that each word appears to all the other words in the corpus. The matrix for our sentence is shown in table 2. The rows of the matrix can give us an initialization of word vectors.

I = [0 1 0 1 1 0] - Love = [1 0 1 0 0 0]
Sea = [0 1 0 0 1 0] - E = [1 0 1 0 0 0]
I like it = [1 0 0 0 0 0] - Fish = [0 0 0 0 1 0]

The matrix of table 2, although simple    supply us

useful information. For example, the words "love" and "like" contain both values 1 in their counts with names Sea and fish.

With a data set larger than a single sentence, you can imagine that this resemblance will become clearer than "like" and "love" and other synonyms will begin to have similar word vectors due to use in similar contexts.

|       | I | Love | Sea | And | Like | Fish |
|-------|---|------|-----|-----|------|------|
| I     | 0 | 1    | 0   | 1   | 1    | 0    |
| Love  | 1 | 0    | 1   | 0   | 0    | 0    |
| Sea   | 0 | 1    | 0   | 0   | 1    | 0    |
| And   | 1 | 0    | 1   | 0   | 0    | 0    |
| Like  | 1 | 0    | 0   | 0   | 0    | 0    |
| Fish  | 0 | 1    | 0   | 0   | 1    | 0    |

**Table 2**. *Cooccurrence matrix of the sentence "I love sea and like fish."*

Now, although this is a good starting point, we note that the size of each word will increase linearly with the size of the corpus. If we had a million words, we would have an array of millions of millions that would be extremely sparse, with lots of 0. There has been much progress in finding the best ways to represent these word vectors. The most famous is Word2Vec.

Word2Vec. Basically we want to store as much as possible in the word vector while maintaining dimensionality on a manageable scale: from 25 to 1000. Word2Vec works on the idea of predicting the surrounding words of each word. Let's take our previous sentence: "I love the sea and I like fish".

Let's look at the first 3 words of this sentence, adopting 3 as the window size 'm'. Now, we will take the center of the word "love" and take the words that come before and after it. This can be achieved by maximizing / optimizing the function (1), which attempts to maximize, given the current central word, the log probability of any context word.

The cost function described above adds the log probabilities of "I" conditional on the probability of "Love" and the probability of "Sea" conditions conditioned with respect to the probability of "Love", being "Love" the central word in both cases. The T parameter represents the number of training sentences The probability of log has the formula (2).

Where Vc is the vector word of the central word. $u_o$ is the word vector representation when it is the central

word, and $u_w$ is the word vector representation when used as an external word[36].

The above cost function adds the log probabilities of "I" conditioned respect to probability of "Love" and the log probability of "Sea" conditioned respect to probability of "Love", being "Love" the center word in both cases. The parameter T represents the number of training sentences.

The log probability has the formula (2).

$$J(\theta) = \frac{1}{T}\sum_{t=1}^{T}\sum_{-m \le j \le m, j \ne 0} \log p(w_{t+j}|w_t) \quad (1)$$

Where $V_c$ is the word vector of center word. $u_o$ is the word vector representation when it is the center word, and $u_w$ is the word vector representation when it is used as the outer word[36].

The vectors are trained with *Stochastic Gradient Descent*. Synthesizing: *Word2Vec* find vector representations of different words by maximizing the log probability of context words, given a center word,

$$\log p(o|c) = \log \frac{\exp\left(u_o^T v_c\right)}{\sum_{w=1}^{W}\exp\left(u_w^T v_c\right)} \quad (2)$$

and by modifying the vectors with SGD. A great contribution of Word2Vec was the *emergence of linear relationships between different word vectors*. The word vectors, with training, incredibly *captures evident grammatical and semantic concepts*.

Another method of initializing the word vector is GloVe which combines the co-occurrence matrices with Word2Vec.

**Recurrent Neural Networks (RNN)** [16] [21] [23].

Now we ask how the vectors of the word fit into Recurrent Neural Networks (RNN).

RNN are important for many NLP tasks. They are able to effective use data from the previous time step. In figure 1 it is shown a part of RNN.

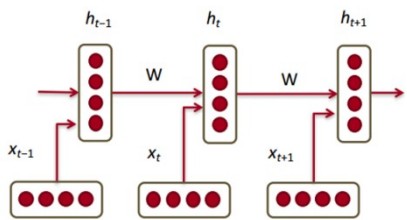

**Figure 2.** Recurrent Neural Network

$$h_t = \sigma \left( W^{(hh)} h_{t-1} + W^{(hx)} x_{[t]} \right) \quad (3)$$

The word vectors are in the bottom and each vector ($x_{t-1}$, $x_t$, $x_{t+1}$) has a hidden state vector at the same time step ($h_{t-1}$, $h_t$, $h_{t+1}$). We call this a module. The hidden state in each module of the RNN is a *function* of both the word vector and the hidden state vector at the previous time step.

In (3) a weight matrix $W^{hx}$ are going to multiply with our input, a recurrent weight matrix $W^{hh}$ which is multiplied with the hidden state vector at the *previous* time step.

These recurrent weight matrices are the *same* across all time steps. **RNN** is very different from a traditional 2 layer NN where we normally have a distinct W matrix for each layer (W1 and W2).

**The Deep Neural Networks**. Neural networks with multiple layers of neurons, accelerated in the calculation with the use of GPUs, have recently seen enormous successes in many fields. They have passed the previous state of the art in speech recognition, object recognition, images, linguistic modeling and translation.

In order to introduce the deep learning symbolism we show in figure 3 a *deep neural network*. This  has three inputs ($i_1$, $i_2$, $i_3$), a first hidden layer ("**A**") with four neurons, a second hidden layer (" **B** ") with five neurons and two outputs ($O_1$, $O_2$), and named by the sequence:  **3-4-5-2.** The network requires a total of (**3** * **4**) weights + 4 *bias* + (**4** * 5)

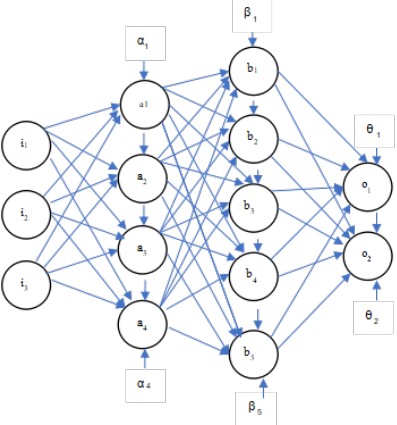

Figure 3. Example of Deep Neural Network 3452

weights + 5 bias + (**5** * 2) weights + 2 bias = **42** weights and 11 bias.

The activation functions are: the *hyperbolic tangent* for    the outputs of the hidden levels and the *softmax* for the output of the network.
The calculation of the feed forward is given by the formulas:
$A_i$ =  tanh ($i_1 p_{1i}$ + $i_2 p_{2i}$ + $i_3 p_{3i}$ + $\alpha_i$), *I hidden layer,*
$B_i$=tanh ($A_1 p_{1i}$ + $A_2 p_{2i}$ + $A_3 p_{3i}$ + $\alpha_i$) *II hidden layer*,
$O_i$ =softmax ($B_1 p_{1i}$ + $B_2 p_{2i}$ + $B_3 p_{3i}$ + $B_3 p_{3i}$ + $\beta_i$), *Output.*
The **standard training of Deep NN** uses back-propagation, but is more difficult than training of NN with a single hidden layer. This fact is the biggest obstacle to the introduction of two or more hidden layers: the back propagation sometime does not give optimal results

### III.   **Deep Learning to Train a Chatbo**t

A chatbot has to be able to determine the best response for any received message, understand the intentions of the message sender, the type of response message required and its grammatically and lexically correct form. Current chatbots are in difficulty facing these tasks and overcome it by introducing Deep L NN. In this case they use some variants of the sequence to sequence (*Seq2Seq*) model[30][31][33]  .
DNNs work well when large labeled training sets are available, but not to map sequences to sequences.
In this paper we were inspired to ***Sutskever method*** by using a multilayered Long Short-Term Memory (LSTM) [1]  to map the input sequence to a vector of fixed dimension; then a deep LSTM to decode target sequence from the vector. The ***seq2seq model*** is constituted by an *encoder RNN* and a *decoder RNN*. Encoder's task is to encapsulate the input text in a fixed representation, while Decoder's task is to derive from it a variable length text that best responds to it.
RNN contains a number of hidden state vectors, and the final hidden state vector of the encoder RNN contains an accurate representation of the whole input text. In the decoder RNN the first cell's task is to take in the vector representation a variable length texts that is the most appropriate for the output response. Mathematically speaking, there are computed probabilities for each words in the vocabulary, and it is chosen the argmax of the values:

$$p(y_1, \ldots, y_{T'}|x_1, \ldots, x_T) = \prod_{t=1}^{T'} p(y_t|v, y_1, \ldots, y_{t-})$$

***Dataset Selection*** is fundamental to train the model. For Seq2Seq models, we need a large number of conversation logs. From a high level, this encoder decoder network needs to be able to understand the type of responses (decoder outputs) that are expected for every query (encoder inputs). For this they are **available various datasets** like *Ubuntu corpus*, *Microsoft's Social Media Conversation Corpus, or Cornell Movie Dialog Corpus*. But the public data set contains a number of data not always useful. Then it may be **better to generate their proper word vectors**. To this aim we can use the before seen approach of a *Word2Vec model.*

*Speech Recognition* and *Speech Synthesis* are indispensable steps for completing user interaction with chatbot, These are very developed areas with important results[14 17 22 24 26 27 29].

## IV. Conclusion

The paper addresses the implementation of an Artificial Intelligent Chatbot capable of talking to man. The system has to understand human language and learn from interactions, increasing its knowledge. Assistive software robots is important beacause increases quality of life by providing assistence and companionship and supplies clear knowledge about the AI, which need people to clearly understand how the AI machine behaves , since the algorithms that drive it are not at all. To understand how an AI System works, we can show how a Chatbot may supply knowledge to the people. We conceived a Chatbot with integrated intelligent software, and capable to learn from the experience. The means to achieve this is constituted by machine learning capability. Recently theoretical results suggest to adopt machine learning algorithms with a deep architectures, in order to learn the kind of complicated functions with high-level abstractions , like in natural language. Fields involved in design such a Bot are Machine Learning, Speech Recognition, NLP, and Chatbot Training that can be realized with net platforms. The article, after framing the various tools, analyzed the understanding of natural language aimed to chatbot training. We have also analyzed Deep Learning, Recurrent Neural Networks, Seq2Seq and Sec2Vet language models. The

Dataset Selection is conceived by looking at datasets in net. This constitutes an approach to building of our intelligent Chatbot using net platform.

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
