# OpenReview forum: "Chatbot: a Key for Understanding AI Revolution"
_icaps-conference.org/ICAPS/2019/Workshop/XAIP — Submitted to XAIP 2019_

### Official Review · AnonReviewer2 · 2019-04-24
**Vague paper on chatbots, with technical content relating mostly to DNN and language understanding**

**Rating:** 2
**Confidence:** 2

**Review:**

I find it a bit difficult to make sense of this paper, and why it has been submitted to XAIP.

I can agree that chatbots, as a general concept, may be of great interest to XAI in shaping the understanding process as a dialogue.

But I find it hard to see what the present paper contributes, along these lines, to XAI in general and to XAIP in particular.

The paper basically consists of two parts, the first of which outlines chatbots in a very abstract and vague way, while the second part specifies details of anNN approach to language understanding. The second part is misplaced in XAIP. The first part might be suited to a position/challenge presentation of sorts, but I find it too vague to get a take-away from it.

Overall, I don't get much from this paper beyond the claim that chatbots can be useful in XAI.

I hence lean to reject. It might be suitable to invite the author to a discussion panel or some such, if such a thing is planed in the workshop.

---

### Official Review · AnonReviewer1 · 2019-05-09
**Unformatted and in need a heavy rewrites.**

**Rating:** 1
**Confidence:** 2

**Review:**

Major, Major, Major: This does not follow the formatting guidelines and trying to read 7 pages of this is arduous.

The abstract is suppose to tell me about the paper, this sounds like an upsell for this ChatBot program.

XAI is not "understandable AI".

Is '37' hear a footnote, or a citation?  You utilize a different form a citation later in the paper (see paragraph to the right).  Follow format guidelines.

You have a footnote in what appears to be a citation? I'm super confused now.

Okay, I'm getting a basic understanding of where this is going in the concept of setting up the Introduction, but this needs a good extra pass or two on the writing to clean it all up. This feels like a first draft.

Page 3 has a verbatim paragraph on XAI from the intro again?

Page 4, don't word-wrap around equations and put figures to top or bottom of the page.

Why does a section heading have footnotes? (RNN)

Page 6: Equation overruns your margins.

I'm surprised to find the Conclusion already, I thought this was still background/motivation. I'm unsure in the reading what the novel concept / introduction of this paper does in comparison to describing others work?

Minor:
  Page 2, bottom left side, "raccomeanations -> recommendations", "coarse -> course", "abitude" -> "attitude"

---

### Decision · Program_Chairs · 2019-05-15

**Decision:**

Reject

**Comment:**

The reviewers agree that the paper is not suited to the workshop. We suggest you revise the paper and send it to a different, more ML oriented, XAI workshop. Good luck with future iterations of your work.